# Examination of food consumption in United States adults and the prevalence of inflammatory bowel disease using National Health Interview Survey 2015

**Moon K. Han** [1]*, **Raeda Anderson**[2], **Emilie Viennois**[1], **Didier Merlin**[1,3]

**1** Institute for Biomedical Sciences, Center for Diagnostics and Therapeutics, Center for Inflammation, Immunity and Infection, Digestive Disease Research Group, Georgia State University, Atlanta, Georgia, United States of America, **2** Department of Research and Engagement, Georgia State University, Atlanta, Georgia, United States of America, **3** Atlanta Veterans Affairs Medical Center, Decatur, Georgia, United States of America

* moonkwon.han@gmail.com

**Data Availability Statement:** The data underlying the results presented in the study are available

## Abstract

Various diets and food components have been implicated as one of the environmental factors associated with inflammatory bowel disease (IBD). Patients are often recommended nutritional guidelines to manage disease symptoms. However, the current food consumption pattern of US adults with IBD that are nationally representative is unclear. A secondary analysis of National Health Interview Survey 2015 was performed to characterize the estimated US adults with IBD and their food intake and consumption frequency using bivariate and multivariate logistic regression. Fries were consumed by a greater number of people with IBD. IBD population drank less 100% fruit juice and ate more cheese and cookies than non-IBD population. Intake of fries (OR 1.60, 95% CI 1.14–2.25) and sports and energy drinks (OR 1.46, 95% CI 1.07–1.97) and more frequent drinking of regular soda were significantly associated with the likelihood of having been told one have IBD, while popcorn (OR 0.73, 95% CI 0.548–0.971) and milk (OR 0.70, 95% CI 0.497–0.998) were associated with smaller odds, adjusting for covariates. Foods typically labeled as junk food were positively associated with IBD. Nonetheless, of the assessed 26 foods, we found eating patterns between IBD and non-IBD population to be mostly analogous. It is unclear whether the results reflect potential change in food intake in IBD population long before the survey interview. Understanding the role of food intake in IBD risk/prevalence would benefit from identifying other environmental factors (i.e. food desert), food processing (i.e. frying), and potential bioactive food components that can induce intestinal inflammation that can increase the individual's susceptibility to IBD.

from https://www.cdc.gov/nchs/nhis/nhis_2015_data_release.htm.

**Funding:** D.M is funded from National Institutes of Health of Diabetes and Digestive and Kidney Diseases (R01-DK-116306). D.M. is a recipient of a Senior Research Career Scientist Award from the Department of Veterans Affairs. E.V. is a recipient of a Career Development Award from the Crohn's and Colitis Foundation. E. Viennois and D. Merlin provided guidance in study approach, manuscript preparation and editing.

**Competing interests:** The authors have declared that no competing interests exist.

## Introduction

Inflammatory bowel disease (IBD) is a chronic condition of the gastrointestinal tract of which approximately 3.1 million adults in the United States are affected by the disease, according to the National Health Interview Survey (NHIS) from 2015 [1]. The two most common conditions of IBD are ulcerative colitis (UC) and Crohn's disease (CD). While the disease is prevalent across all ages, disease onset peaks at early adulthood [2–4] and persists throughout life with interim remission with surgery being the last resort upon ineffectiveness of available therapy. Relapsing symptoms of diarrhea, abdominal pain, intestinal bleeding, and malnutrition and weight loss due to nutrient malabsorption [5,6] can be increasingly debilitating, potentially reducing the quality of life and the ability to thrive physically [5] and socially [7]. A definitive cause of IBD is unknown. However, IBD has been strongly associated with a genetic predisposition, gut microbiota composition, altered innate and adaptive immune responses [8–11]. Just as the etiology of IBD is multi-faceted, studies over the years have unveiled environmental factors such as diet, lifestyle, and social factors as part of a critical component contributing to the disease risk [12,13]. Regardless of different attributable risks associated with the disease, conventional approach to treat IBD primarily targets inflammation driven by T-cell mediated cytokine production and other pro-inflammatory effectors [14,15], as there is no known cure, to prevent relapses and manage the inflammation. Compounded by the chronic inflammation presented by the disease, a patient with clinically diagnosed IBD is at an increased risk for developing colon cancer, third leading cancer for new cases and related deaths for both men and women in US [16], deepening the burden to public health care and associated costs [17,18].

The literature suggests an association of diet and nutrients as a potential risk factor of IBD [19–21]. Due to symptoms inflicted by IBD and compromised functions of the small intestine for proper nutrient absorption [22–24], diet and nutrient recommendation are provided to remedy the nutrient deficiency and related morbidities, such as anemia due to iron deficiency, experienced by IBD patients. Indeed, according to Crohn's & Colitis Foundation, certain fluids, selective sources of fiber and whole grains, fruits and vegetables, proteins, and calcium are recommended while avoiding certain foods within the same food group [25] for disease management. A systemic review of the literature has found high consumption of total fat, polyunsaturated fatty acids, omega-6 fatty acids, saturated fats, and meats to be associated with increased risk for IBD [19]. Furthermore, a greater intake of fruits and dietary fiber were associated with decreased risk for Crohn's disease, but not for ulcerative colitis [19]. Cross-sectional studies on IBD patients have found evidence of a positive association of meat consumption and disease development [26,27] but also saw nutrient deficiency [26] and diet change as a result of the disease [28,29]. With the urbanization and globalization of economy, increasing incidences of inflammatory diseases, such as autoimmune disease and obesity-related health conditions as well as IBD [30–34], is being linked to the adoption of westernized lifestyle and diet [19,35–37]. However, a definitive role of various food intake in disease etiology or symptom management still needs much elucidation.

To our current knowledge, a relationship between food consumption and prevalence of IBD in US adult population has only been reported through small sample size cohort studies and literature meta-analysis. Here, we used a cross-sectional national health survey data to capture a current reflection of different food intake patterns in population who were once told they had/have IBD compared to those who were not diagnosed of IBD. Albeit the nature of questionnaire not being specific to IBD, we believe the use of NHIS 2015 is both appropriate and pivotal in determining the aforementioned relationship due to the ascertainment of IBD status and food intake (through Cancer Control Module) in randomly selected sample population in this complex survey design for 2015.

## Methods

### Survey and datasets

The analysis was performed using 2015 National Health Interview Survey (NHIS), which has the quinquennial Cancer Control Module (CCM) as a supplemental assessment. NHIS is a cross-sectional household survey conducted yearly since 1960. Survey participants are interviewed face-to-face with computer assistance. Questionnaire consists of six sections: Household, Family, Person, Sample Child, Injury Episode, and Sample Adult. For the current analysis, data from Person file, Sample Adult file, and Sample Adult Cancer file from CCM were primarily used in addition to Imputed Income files for the analysis of family income to poverty threshold ratio. All responses are self-reported. Survey weights derived from the estimates of 2010 census-based population [38] were applied for each appropriate analysis.

### Study variables

**Demographic variables.** Sample adult survey participants consisted of non-institutionalized adults between the ages of $\geq$ 18 to $\leq$ 85 who answered NHIS survey and CCM concurrently. Participants who self-reported their gender as man or woman were analyzed. Ethnicity was categorized to either Hispanic or Non-Hispanic. Race was categorized as follows: White only; Black or African-American; American-Indian or Alaska Natives; Asian only; multiple races; race unknown. Age groups analyzed were 18–24, 25–34, 35–44, 45 and 54, 55–64, 65–74, and 75–85 years old.

Highest education level completed by the adult participants were categorized as follows: kindergarten or never attended; primary school only (Grades 1–5); junior high school only (Grades 6–8); some high school (Did not graduate); high school or general education development; some college (did not graduate); 2-Year college; 4-Year college; advanced and terminal degrees (Masters, Doctorates). Family income to poverty threshold ratio (IPR) was obtained following the imputed income data analysis guideline provided by CDC [39]. To isolate and analyze only the adults and their corresponding IPR, a dummy variable was created, and subsequently categorized based on the IPR as poor (Less than 100% of poverty threshold), near poor (100% to less than 200% of poverty threshold), or not poor (200% of poverty threshold or greater). Region of the participants' residence at the time of the interview was also included. Poverty status was used as a proxy for the socioeconomic factor.

**Lifestyle variables.** Participant body mass index (BMI, kg/m$^2$) information was categorized into the following groups: underweight (BMI < 18.5); healthy (BMI between 18.5 - < 25); overweight (BMI between 25 - <30); obese (BMI of $\geq$ 30). Participant was categorized as having ever smoked, if he/she self-reportedly "smoked at least 100 cigarettes in his/her entire life"; "ever used smokeless tobacco products even one time"; "ever smoked a regular cigar, cigarillo, or a little filtered cigar even one time"; "ever smoked a pipe filled with tobacco-either a regular pipe, water pipe, or hookah even one time". To be considered having never smoked, the participant must have answered "No" to all four questions. Alcohol use information was subdivided into two categories to identify user status and consumption status. User status was categorized as the following based on their self-reported drinking habits: abstainer (Alcohol consumption is < 12 times in lifetime); former (Alcohol consumption > 12 times in lifetime but none in past year); current (Alcohol consumption is > 12 in lifetime and consumed at least 1 drink in the past year). Alcohol consumption status was categorized as the following: abstainer (consumed < 12 times in lifetime); infrequent (consumed $\leq$ 12 times a year); regular or light (consumed > 12 times a year but $\leq$ 3 a week in the past year); moderate (consumed 3–14 times a week for men, 3–7 times a week for women); heavy (consumed > 14 times a week for men; > 7 times a week for women).

**Food variables.** From CCM, the following food items were assessed: cereal (hot or cold), popcorn, brown rice (and other whole grains), whole grain bread, fries (or any other fried potatoes), salad (including green leaf or lettuce of any kind), 100% pure fruit juice, vegetables, non-fried potato, pizza, fruits, tomato sauce, salsa (made with tomatoes), beans, milk (from cow), cheese (excluding cheese from pizza), ice cream (or frozen desserts), processed meat, red meat, cookies (including pies, cakes, brownies), donuts (including pastries and muffins), candy (including chocolate), sports and energy drinks, regular soda (or pop), coffee or tea (sweetened with sugar or honey), and fruit drinks (sweetened with sugar). For the food intake variables, participants were asked "During the past month, how often did you have/eat [food item]? You can tell me per [day, week, or month]". The purpose of the question is to ascertain the consumption frequency of foods items being evaluated, prompting the participants to give a past 30-day account of intake. Subsequently, the responses to nutrition and diet questions were subdivided into two parts: number of units of consumption (i.e. 3 times) and the consumption rate (i.e. per month). To streamline the analysis, we have converted the values to reflect the monthly consumption. According to the survey, the food consumption recall is based on past 30-day consumption. To create a binary response of whether the participant consumed the food item in the past month (30 days), any participants who self-reported such consumption, regardless of the frequency, was considered having consumed the food item. Likewise, who responded "Never" to the question was considered not having consumed the food item in the past month. Due to the nature of responses based on recalls and self-reports, responses to the food consumption frequency included unusually large values that seem unreasonable from the practical point of view. To address its potential effect on the overall data and per recommendation and methods provided by National Cancer Institute, the maximum frequency value for extreme values in each diet item was applied [40]. Any observations for the food variable exceeding the maximum frequency value allowed were top-coded accordingly to prevent being lost/excluded in the analysis. Summary statistics and modeling are based on this change.

**Outcome variable.** For 2015 NHIS, Crohn's disease/ulcerative colitis was included as one of the health assessments for sample adults. The outcome of IBD were assessed from the response of the following question: "Have you ever been told by a doctor or other health professional that you had Crohn's disease or ulcerative colitis?" From here on forward, participants will be referred to as having IBD or not having IBD (non-IBD).

## Analysis

All analysis was performed using Stata/IC 15 (StataCorp LLC, Texas, USA). Survey weight and the survey design designation to the working dataset were commanded to allow for weighted data analysis. A significance level was set at 0.05 for all tests. To test the difference in proportions of categorical responses, F-statistics was performed to test the null hypothesis of equal proportions. Independence of two categorical variables was tested for the null hypothesis of no association. The estimation of the likelihood of IBD as an outcome with the consumption of individual food item was reported as odds ratio (OR) and 95% confidence interval (95% CI) for the bivariate/multivariate analysis. Regression analysis was controlled for demographic, or lifestyle, or demographic and lifestyle variables. To characterize the prevalence of IBD in those consuming greater than ($>$) or less than or equal to ($\leq$) the average (median, 50$^{th}$ percentile) monthly food intake frequency, a binary dummy variable was created using the median as a cutoff value. Point prevalence of IBD within subpopulation with different consumption behavior was estimated.

## Results

### Characteristics of survey participants (unweighted)

Total survey population was 103,789 (**S1 Table** of **S1.1 Table**). There was a total of 33,672 adults aged between $\geq$ 18 and $\leq$ 85. Among the adults, 44.76% were men and 55.24% were women. The least amount of people participating in the survey were in the age group of 18–24 and 75–85, with 8.58% and 10.94%, respectively. Ethnicity was reported as 83.40% as Non-Hispanic and 16.60% as Hispanic. Most participants were White in racial background (76.71%), followed by Black or African-American (13.88%), Asian (5.89%), multiple race (2.08%), and American Indian or Alaskan Natives (1.16%). Race was unknown for less than half a percent of the participants. Participants were recruited from the regions of south (34.59%), west (27.75%), midwest (21.09%), and northeast (16.57%). Highest education attainment in the survey adult population ranged widely with a greater number of people having completed high school (HS) or received GED (24.82%) or attended some college (19.51%) or completed 4-year college (18.48%). According to the family income to poverty threshold ratio (IPR), 15.86% of the participants were considered poor, 20.85% were considered near poor, and over 63% were considered not poor. While smoking status was unknown for 3.77% of the survey participants, 50% has ever smoked and 46.17% never smoked. When the alcohol user status was assessed, 62.52% were current drinkers and 15.60% were former drinkers, and 20.58% were abstainers. When the alcohol frequency was assessed, 35.63% drank regularly or lightly, 22.52% drank infrequently, 14.66% drank moderately, and 4.99% drank heavily. Alcohol use and the consumption rate was unknown for less than 2% of the survey participants. According to body mass index (BMI), 1.79% was underweight, 32.26% had healthy BMI, 32.80% were overweight, and 29.75% were considered obese. Amongst the survey participants, 454 responders or 1.35%, were ever told by health professionals or medical doctors that they have IBD (ulcerative colitis/Crohn's disease).

### Characteristics of estimated population with IBD

Prevalence of IBD among estimated US adults is 1.28% (95% CI 1.27–1.28; **Fig 1A** and **S1 Table** of **S1.2 Table**). Among estimated population with IBD, women were more likely to have IBD than men (57.41% vs. 42.59%, respectively; p-value 0.0234; **Fig 1B**). Over 87% of IBD population are non-Hispanic in ethnic background (**Fig 1B**). IBD was more prevalent amongst the population whose poverty threshold was at 200% or greater (66.32%) compared to poor (15.73%) or near poor (17.95%) (**Fig 1B**). Adults aged between 18–24 were least likely to have IBD (4.95%), while those between the age of 55–64 years had the greatest number of people with IBD (23.13%) (vs. 18–24: p-value = 0.0009; vs. 25–34: p-value = 0.0659; vs. 35–44: p-value = 0.0028; vs. 45–54: p-value = 0.1848; vs. 65–74: p-value = 0.0557; vs. 75–85: p-value = 0.0003; **Fig 1C**). White (88.18%) were also more likely to have been told that they have IBD than people of other races (**Fig 1D**). Population with IBD were more likely reside in south (38.32%) than in northeast (19.35%), midwest (22.09%), or west (20.25%) (**Fig 1E**). Highest education attained varied with the greatest proportion receiving HS or GED (25.06%) (**Fig 1F**). A greater percentage of people with IBD were current drinkers (58.15%) and tended to drink regularly or lightly (**Fig 2A** and **2B**). Smoking was also very prevalent in IBD population (**Fig 2C**), and the disease was more common in people with healthy and overweight BMI (18.5 to < 30 kg/m², **Fig 2D**).

### Characteristics of food consumption in estimated IBD population

Food items listed in the Diet and Nutrition questionnaire from the CCM were evaluated (**S2 Table**). The assessment of the sample population and an estimated population who have

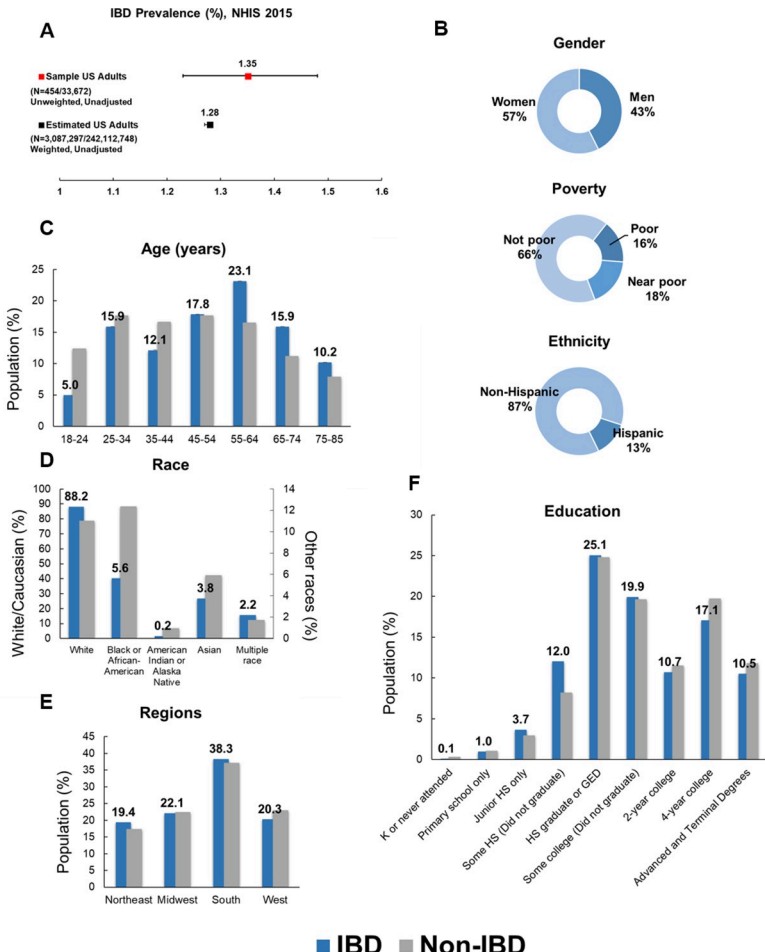

**Fig 1. Demography characteristics of estimated US adult population with inflammatory bowel disease (IBD) from National Health Interview Survey (NHIS), 2015.** (**A**) IBD prevalence (%) in sample US adult survey participants and estimated US adult population. **B**) Distribution of gender, poverty status, and ethnicity shown in percentages. (**C**) Age, (**D**) race, (**E**) regions of residence, and (**F**) education. Values in panels C, D, E, and F reflect those of IBD population. Error bars indicate lower and upper bound of 95% confidence interval. N = (Number of people with IBD/Total number) *Asterisks indicate statistically significant difference (p-value < 0.05) when compared to estimated non-IBD population.

consumed each food item in the past month are displayed in **S3 Table**. Among the estimated US adult population with IBD and without IBD, the number of people (in proportions) with certain food intake in the past 30 days (**Table 1** and **Fig 3**) were similar for both populations in the following food groups: dairy (milk, cheese, pizza, ice cream) and meat (processed meat, red meat) (**Fig 3A**); sweetened food/drinks (cereal, cookies, donuts, coffee or tea, fruit drinks, candy, sports and energy drinks, regular soda, ice cream) (**Fig 3B**); whole wheat grains (popcorn, cereal, brown rice, whole grain bread) (**Fig 3C**); fruit and vegetables (**Fig 3D**). However, we found significantly greater number of people with IBD to eat fries, than the general population without IBD (84.73% vs. 79.75%, F-test = 5.78, p-value 0.0168).

## Relationship of food consumption and IBD

**Food consumption (binary response) and prevalence of IBD.** To determine the likelihood of having IBD based on whether a person has consumed the food item in the past

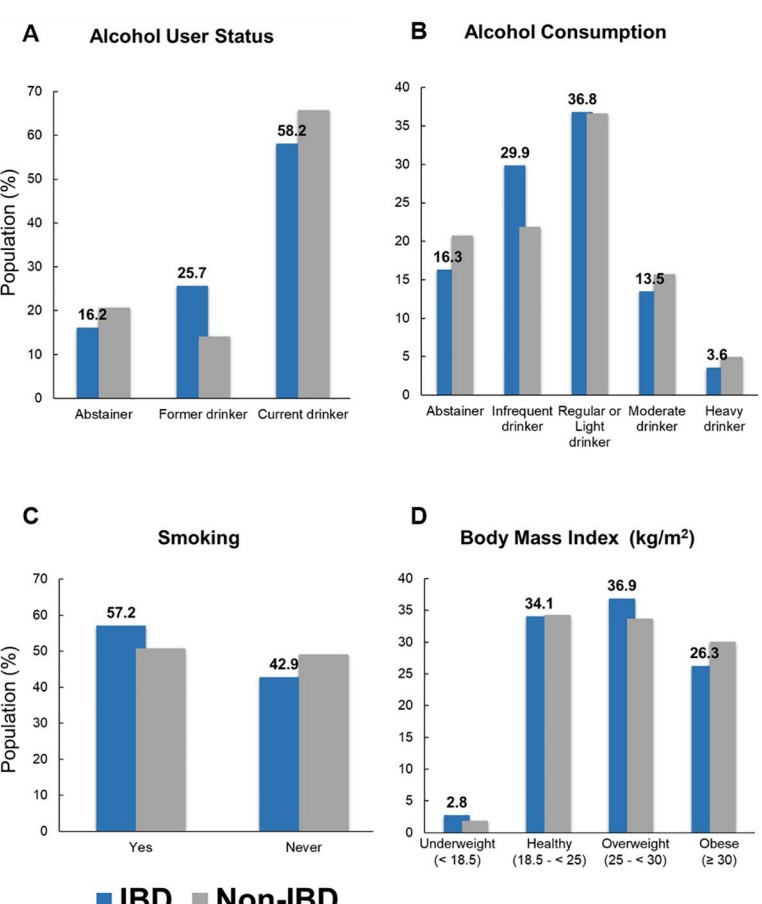

**Fig 2. Lifestyle characteristics of estimated US adult population with inflammatory bowel disease (IBD) from National Health Interview Survey (NHIS), 2015.** (**A**) Alcohol user status, (**B**) alcohol consumption status, (**C**) smoking, and (**D**) Body Mass Index (BMI, kg/m$^2$). Values in panels A, B, C, and D reflect those of IBD population. Error bars indicate lower and upper bound of 95% confidence interval. N = (Number of people with IBD/Total number) *Asterisks indicate statistically significant difference (p-value < 0.05) when compared to estimated non-IBD population. K-Kindergarten; HS-High school; GED -General Education Development.

month, logistic regression on each food item was applied with IBD as an outcome, and the results are depicted in **S4 Table**. According to the analysis on weighted, but unadjusted data of binary response of food consumption, the odds of having IBD was 1.41 times for those who consumed fries in the past month than that of those who did not eat fries (95% CI [1.030–1.929], p-value = 0.032). When adjusted for demographic factors such as age, race, poverty status, gender, ethnicity, and region or the lifestyle factors such as smoking status, alcohol user status, alcohol consumption rate status, and BMI, the odds remained high at 1.63 (95% CI [1.189–2.245], p-value = 0.003) and 1.40 (95% CI [1.022–1.924], p-value = 0.036), respectively, for those who consumed fries in the past month. This observation remained unaltered when both demography and lifestyle factors were accounted for in a full model (OR = 1.63, 95% CI [1.183–2.238], p-value = 0.003). The likelihood of having told that one has IBD in those who drank sports and energy drinks was 1.48 times (95% CI [1.099–1.987], p-value = 0.010) and 1.50 times (95% CI [1.116–2.027], p-value = 0.008) that of the non-consumers adjusting for demographic factors alone or both, respectively.

For those who have reported of popcorn or milk intake, the likelihood of having been told that one has IBD were 0.73 times (95% CI [0.5445–0.9674], p-value = 0.029) and 0.67 times

**Table 1. Comparison of weighted proportions of food item consumption in IBD and non-IBD estimated US population, NHIS 2015[a,b].**

| Food groups[c] | Food items | IBD = Yes | | | IBD = No | | | F-test[f] | p-value |
|---|---|---|---|---|---|---|---|---|---|
| | | N | Percent | 95% CI | N | Percent | 95% CI | | |
| Whole wheat grains | Popcorn | 1,306,270 | 44.55 | (38.04–51.26) | 114,468,924 | 51.21 | (50.37–52.06) | 3.840 | 0.0511 |
| | Cereal (hot or cold)[g] | 2,198,692 | 75.34 | (68.72–80.95) | 160,837,887 | 71.52 | (70.81–72.23) | 1.480 | 0.2247 |
| | Brown rice | 1,375,009 | 46.90 | (40.44–53.46) | 112,253,586 | 50.13 | (49.34–50.92) | 0.930 | 0.3362 |
| | Whole grain bread | 2,179,865 | 74.35 | (69.16–78.93) | 171,970,787 | 76.94 | (76.23–77.63) | 1.080 | 0.2999 |
| Fruits and vegetables | Fries | 2,472,687 | 84.73 | (80.28–88.33) | 178,827,691 | 79.75 | (79.06–80.42) | 5.780 | 0.0168* |
| | Salad (green leafy, lettuce) | 2,551,381 | 87.02 | (82.62–90.43) | 202,636,603 | 90.28 | (89.78–90.77) | 2.660 | 0.1038 |
| | Fruit juices (100% pure fruit juice) | 1,812,407 | 61.74 | (55.13–67.95) | 148,915,123 | 66.36 | (65.54–67.18) | 2.010 | 0.1576 |
| | Vegetables[d] | 2,760,074 | 94.14 | (90.72–96.34) | 213,878,256 | 95.55 | (95.20–95.87) | 1.020 | 0.3142 |
| | Potato (non-fried) | 2,547,714 | 86.89 | (82.28–90.45) | 190,582,800 | 85.10 | (84.54–85.64) | 0.730 | 0.3937 |
| | Pizza (frozen, fast food, homemade) | 2,375,321 | 81.01 | (74.84–85.95) | 158,123,747 | 82.62 | (81.99–83.22) | 0.320 | 0.5691 |
| | Fruits (fresh, frozen, canned) | 2,700,634 | 92.22 | (88.56–94.78) | 209,302,985 | 93.26 | (92.82–93.67) | 0.440 | 0.5077 |
| | Tomato sauce | 2,395,891 | 81.82 | (76.10–86.41) | 185,708,495 | 83.07 | (82.46–83.65) | 0.230 | 0.6351 |
| | Salsa (made with tomatoes) | 1,873,306 | 63.89 | (57.36–69.95) | 146,030,331 | 65.25 | (64.43–66.06) | 0.170 | 0.6792 |
| | Beans | 2,247,672 | 76.66 | (70.72–81.70) | 173,734,461 | 77.62 | (76.93–78.29) | 0.120 | 0.7334 |
| Dairy | Milk (cow milk, any type) | 2,087,082 | 71.29 | (65.06–76.81) | 170,380,909 | 75.83 | (75.10–76.55) | 2.180 | 0.1404 |
| | Cheese (excludes cheese on pizza) | 2,740,059 | 93.57 | (90.14–95.86) | 206,708,117 | 92.41 | (92.01–92.80) | 0.660 | 0.4184 |
| | Pizza (frozen, fast food, homemade)[g] | 2,375,321 | 81.01 | (74.84–85.95) | 158,123,747 | 82.62 | (74.84–85.95) | 0.320 | 0.5691 |
| | Ice cream (frozen desserts)[g] | 2,100,989 | 71.66 | (65.61–77.01) | 159,984,495 | 71.59 | (70.83–72.34) | 0.000 | 0.9815 |
| Meat | Processed meat | 2,343,122 | 80.02 | (74.79–84.38) | 170,564,974 | 76.25 | (75.52–76.97) | 2.370 | 0.1249 |
| | Red meat | 2,696,761 | 92.09 | (88.51–94.62) | 204,199,009 | 91.29 | (90.83–91.73) | 0.280 | 0.5962 |
| Sweetened food/drinks[e] | Cereal (hot or cold)[g] | 2,198,692 | 75.34 | (68.72–80.95) | 160,837,887 | 71.52 | (68.72–80.95) | 1.480 | 0.2247 |
| | Cookies (i.e. cake, pies, brownies) | 2,296,673 | 78.33 | (72.43–83.26) | 168,105,863 | 75.24 | (74.50–75.96) | 1.250 | 0.2638 |
| | Donut (i.e. Danish, pastries, muffins) | 1,720,836 | 58.69 | (52.41–64.70) | 125,706,425 | 56.23 | (55.44–57.02) | 0.610 | 0.4355 |
| | Coffee or tea (sugar or honey added) | 1,552,116 | 52.94 | (46.55–59.22) | 122,830,695 | 54.69 | (53.89–55.49) | 0.300 | 0.5868 |
| | Fruit drinks (sweetened with sugar) | 860,899 | 29.36 | (24.44–34.82) | 62,814,773 | 27.98 | (27.17–28.80) | 0.280 | 0.5956 |
| | Candy (i.e. chocolates) | 2,266,853 | 77.31 | (71.39–82.31) | 170,097,045 | 76.12 | (75.41–76.82) | 0.170 | 0.6765 |
| | Sports and energy drinks | 865,600 | 29.52 | (24.47–35.14) | 63,591,743 | 28.30 | (27.56–29.05) | 0.200 | 0.6544 |
| | Regular soda or pop | 1,632,312 | 55.54 | (48.68–62.19) | 125,886,667 | 56.02 | (55.14–56.89) | 0.020 | 0.8899 |
| | Ice cream (frozen desserts)[g] | 2,100,989 | 71.66 | (65.61–77.01) | 159,984,495 | 71.59 | (65.61–77.01) | 0.000 | 0.9815 |

[a]Weighted using sample weight [wtfa_sa] for cross-tabulation with IBD as outcome; Data source: Sample Adult Cancer file from 2015 NHIS Data release source (https://www.cdc.gov/nchs/nhis/nhis_2015_data_release.htm)

[b]Additional details in survey questions can be found in NHIS 2015 Data release website: ftp://ftp.cdc.gov/pub/Health_Statistics/NCHS/Dataset_Documentation/NHIS/2015/cancerxx_layout.pdf

[c]Food groups are based on the relationship previously established according the dietary guidelines. Details can be found on https://epi.grants.cancer.gov/nhanes/dietscreen/relationship.html.

[d]Vegetables other than lettuce salads, potatoes, cooked beans in which participant already answered to in previous questions.

[e]Food items exclude artificially sweetened or sugar-free kinds

[f]Test of proportion, Adjusted Wald test

[g]Food items appear in more than one food groups: Pizza, Ice cream, Cereal

*Statistically significant; Below the significance level of 0.05

(95% CI [0.4844–0.9403], p-value = 0.020), respectively, to that of those who did not consume the same foods in the past 30 days (S5 Table), when all the food items were included in a weighted but unadjusted full model. For those reported eating cereal or fries, the likelihood of having been told that one has IBD were 1.50 times (95% CI [1.0286–2.1811], p-value = 0.035)

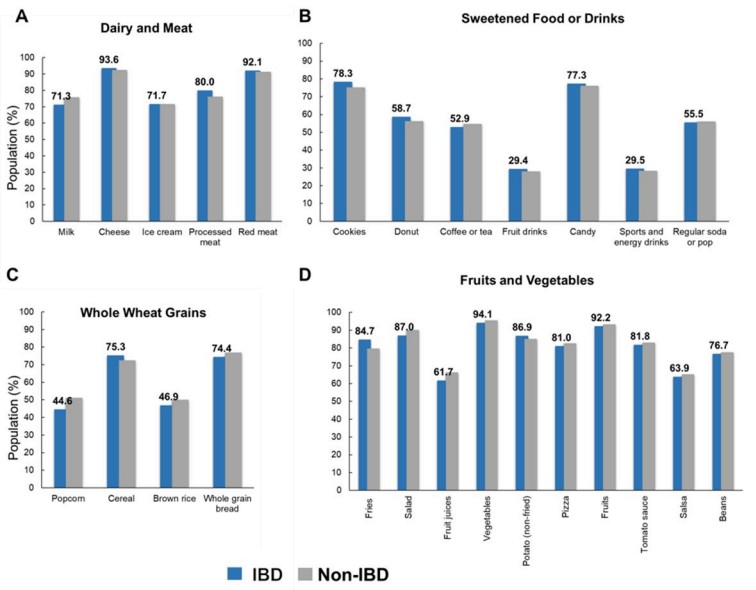

**Fig 3. Comparison of weighted proportion of food intake between estimated US population with IBD and without IBD, NHIS 2015.** Values in panels A, B, C, and D reflect those of IBD population. *Asterisks indicate statistically significant difference (p-value < 0.05) when compared to estimated non-IBD population.

and 1.48 times (95% CI [1.0468–2.0810], p-value = 0.026), respectively, that of those who have never consumed the same food item in the past month. The odds remained at similar values even after adjusting for demography and/or lifestyle variables (**S5 Table**). Interestingly, sports and energy drinks consumption was associated with higher odds of having been told of IBD diagnosis by 43–46% only when adjusted for demography or both all in one model. Significance in the effect of cereal consumption on higher likelihood of having been told of IBD diagnosis was supported only when adjusted for lifestyle predictors.

**Monthly consumption rate and prevalence of IBD.** To determine the odds of ever having been told that one has IBD by a medical or health professional and its association to the food consumption pattern, we fitted the monthly consumption responses (number of times per month, continuous variable) of each individual food items to a logistic regression model with IBD as an outcome variable (**Table 2**). In a weighted, unadjusted analysis, we found a significant association between IBD and consumption of cheese (OR = 1.006, 95% CI [1.0021–1.0104], p-value = 0.003), ice cream (OR = 1.011, 95% CI [1.0022–1.0203], p-value = 0.015), and regular soda (OR = 1.006, 95% CI [1.00967–1.00969], p-value = 0.003). Increasing the frequency of eating fries, cheese, and regular soda was significantly associated with IBD when adjusted for demography (**Table 2**). The odds of having been told of IBD diagnosis was significantly greater in those consuming ice cream in an unadjusted (OR = 1.011, 95% CI [1.0022–1.0203], p-value 0.015) and lifestyle adjusted (OR = 1.011, 95% CI [1.0021–1.0201], p-value 0.016) model. Increasing the intake of fries was significantly associated with greater odds of having been told of IBD diagnosis (OR = 1.011, 95% CI [1.003–1.019], p-value = 0.005, adjusted for both demography and lifestyle). The observation was similar for those who consumed cheese (OR = 1.008, 95% CI [1.004–1.012], p-value<0.001) and regular soda (OR = 1.007, 95% CI [1.003–1.011], p-value = 0.001). Inclusion of all food items in one model (**S6 Table**), in weighted unadjusted, or adjusted for demography and/or lifestyle, has found that increasing monthly intake of cheese and regular soda was significantly associated with greater odds of having been told of IBD diagnosis.

**Table 2. Association (OR[h]) of food consumption frequency and IBD in estimated US population, NHIS 2015[a,b].**

| Food groups[c] | Food items (the order is slightly different) | Weighted, Unadjusted | | | Weighted, Adjusted for Demography[f] | | | Weighted, Adjusted for Lifestyle[g] | | | Weighted, Adjusted for Demography and Lifestyle[f,g] | | |
|---|---|---|---|---|---|---|---|---|---|---|---|---|---|
| | | OR | p-value | 95% CI | OR | p-value | 95% CI | OR | p-value | 95% CI | OR | p-value | 95% CI |
| Whole wheat grains | Popcorn | 0.980 | 0.943 | (0.9472–1.0043) | 0.973 | 0.072 | (0.9467–1.0024) | 0.977 | 0.115 | (0.9487–1.0058) | 0.976 | 0.094 | (0.9479–1.0042) |
| | Cereal (hot or cold) | 1.002 | 0.524 | (0.9955–1.0080) | 1.010 | 0.789 | (0.9939–1.0081) | 1.002 | 0.548 | (0.9953–1.0089) | 1.001 | 0.807 | (0.9937–1.0082) |
| | Brown rice | 0.993 | 0.430 | (0.9746–1.0110) | 1.000 | 0.973 | (0.9835–1.0162) | 0.993 | 0.430 | (0.9744–1.0111) | 1.000 | 0.959 | (0.9831–1.0163) |
| | Whole grain bread | 0.998 | 0.647 | (0.9914–1.0050) | 0.997 | 0.423 | (0.9897–1.0000) | 0.998 | 0.643 | (0.9914–1.0054) | 0.997 | 0.412 | (0.9896–1.0043) |
| Fruits and vegetables | Fries | 1.008 | 0.103 | (0.9985–1.0167) | 1.011 | 0.004* | (1.0034–1.0184) | 1.007 | 0.124 | (0.9980–1.0166) | 1.011 | 0.005* | (1.0033–1.0186) |
| | Salad (green leafy, lettuce) | 0.997 | 0.610 | (0.9869–1.0077) | 0.997 | 0.564 | (0.9858–1.0078) | 0.998 | 0.683 | (0.9876–1.0082) | 0.997 | 0.603 | (0.9861–1.0082) |
| | Fruit juices (100% pure fruit juice) | 0.996 | 0.345 | (0.9873–1.0040) | 0.996 | 0.384 | (0.9877–1.0048) | 0.996 | 0.366 | (0.9872–1.0048) | 0.996 | 0.411 | (0.9877–1.0051) |
| | Vegetables[d] | 0.994 | 0.101 | (0.9878–1.0010) | 0.993 | 0.060 | (0.9857–1.0003) | 0.995 | 0.126 | (0.9884–1.0015) | 0.993 | 0.075 | (0.9862–1.0007) |
| | Potato (non-fried) | 1.010 | 0.054 | (0.9998–1.0207) | 1.006 | 0.299 | (0.9946–1.0178) | 1.011 | 0.031 | (1.0010–1.0212) | 1.007 | 0.203 | (0.9961–1.0184) |
| | Fruits (fresh, frozen, canned) | 0.998 | 0.463 | (0.9924–1.0000) | 0.998 | 0.402 | (0.9918–1.0033) | 0.998 | 0.495 | (0.9924–1.0037) | 0.998 | 0.419 | (0.9916–1.0035) |
| | Pizza (frozen, fast food, homemade) | 0.978 | 0.308 | (0.9359–1.0211) | 0.995 | 0.791 | (0.9569–1.0341) | 0.981 | 0.364 | (0.9422–1.0222) | 0.999 | 0.964 | (0.9640–1.0357) |
| | Tomato sauce | 1.007 | 0.455 | (0.9894–1.0240) | 1.009 | 0.278 | (0.9925–1.0260) | 1.007 | 0.426 | (0.9900–1.0241) | 1.010 | 0.256 | (0.9931–1.0264) |
| | Salsa (made with tomatoes) | 0.990 | 0.178 | (0.9752–1.0040) | 0.996 | 0.639 | (0.9818–1.0113) | 0.991 | 0.213 | (0.9761–1.0054) | 0.997 | 0.647 | (0.9820–1.0114) |
| | Beans | 0.995 | 0.420 | (0.9835–1.0069) | 0.997 | 0.628 | (0.9840–1.0098) | 0.996 | 0.480 | (0.9839–1.0077) | 0.997 | 0.665 | (0.9841–1.0103) |
| Dairy | Milk (cow milk, any type) | 0.997 | 0.463 | (0.9887–1.0052) | 0.996 | 0.385 | (0.9879–1.0047) | 0.997 | 0.454 | (0.9883–1.0053) | 0.996 | 0.388 | (0.9876–1.0049) |
| | Cheese (excludes cheese on pizza) | 1.006 | 0.003* | (1.0021–1.0104) | 1.007 | <0.001* | (1.0032–1.0113) | 1.007 | 0.001* | (1.0027–1.0107) | 1.008 | <0.001* | (1.0038–1.0115) |
| | Pizza (frozen, fast food, homemade) | 0.978 | 0.308 | (0.9359–1.0211) | 0.995 | 0.791 | (0.9569–1.0341) | 0.981 | 0.364 | (0.9422–1.0222) | 0.999 | 0.964 | (0.9640–1.0357) |
| | Ice cream (frozen desserts) | 1.011 | 0.015* | (1.0022–1.0203) | 1.010 | 0.069 | (0.9993–1.0201) | 1.011 | 0.016* | (1.0021–1.0201) | 1.005 | 0.070 | (0.9992–1.0199) |
| Meat | Processed meat | 1.001 | 0.887 | (0.9904–1.0112) | 1.002 | 0.680 | (0.9921–1.0123) | 1.002 | 0.712 | (0.9919–1.0120) | 1.003 | 0.496 | (0.9937–1.0130) |
| | Red meat | 0.997 | 0.566 | (0.9878–1.0067) | 0.999 | 0.870 | (0.9904–1.0082) | 0.998 | 0.668 | (0.9889–1.0072) | 1.000 | 0.987 | (0.9916–1.0086) |
| Sweetened food/drinks[e] | Cereal (hot or cold) | 1.002 | 0.524 | (0.9955–1.0080) | 1.010 | 0.789 | (0.9939–1.0081) | 1.002 | 0.548 | (0.9953–1.0089) | 1.001 | 0.807 | (0.9937–1.0082) |
| | Cookies (i.e cake, pies, brownies) | 1.008 | 0.064 | (0.9996–1.0159) | 1.005 | 0.275 | (0.9959–1.0146) | 1.008 | 0.076 | (0.9922–1.0160) | 1.005 | 0.296 | (0.9956–1.0147) |
| | Donut (i.e. Danish, pastries, muffins) | 1.007 | 0.283 | (0.9940–1.0208) | 1.007 | 0.312 | (0.9935–1.0205) | 1.007 | 0.307 | (0.9935–1.0208) | 1.007 | 0.329 | (0.9932–1.0205) |
| | Candy (i.e chocolates) | 1.006 | 0.077 | (0.9990–1.0119) | 1.005 | 0.171 | (0.9980–1.0112) | 1.006 | 0.063 | (0.9997–1.0119) | 1.007 | 0.155 | (0.9982–1.0112) |
| | Sports and energy drinks | 0.995 | 0.458 | (0.9832–1.0077) | 1.002 | 0.608 | (0.9930–1.0120) | 0.996 | 0.523 | (0.9840–1.0083) | 1.003 | 0.538 | (0.9937–1.0123) |

*(Continued)*

Table 2. (Continued)

| Food groups[c] | Food items (the order is slightly different) | Weighted, Unadjusted | | | Weighted, Adjusted for Demography[f] | | | Weighted, Adjusted for Lifestyle[g] | | | Weighted, Adjusted for Demography and Lifestyle[f,g] | | |
|---|---|---|---|---|---|---|---|---|---|---|---|---|---|
| | | OR | p-value | 95% CI | OR | p-value | 95% CI | OR | p-value | 95% CI | OR | p-value | 95% CI |
| | Coffee or tea (sugar or honey added) | 1.002 | 0.249 | (0.9984–1.0063) | 1.002 | 0.307 | (0.9982–1.0057) | 1.002 | 0.264 | (0.9982–1.0064) | 1.002 | 0.332 | (0.9981–1.0058) |
| | Fruit drinks (sweetened with sugar) | 0.998 | 0.705 | (0.9872–1.0087) | 1.000 | 0.950 | (0.9908–1.0099) | 0.998 | 0.679 | (0.9870–1.0086) | 1.000 | 0.944 | (0.9908–1.0100) |
| | Regular soda or pop | 1.006 | 0.003* | (1.0097–1.0097) | 1.007 | <0.001* | (1.0030–1.0103) | 1.006 | 0.006* | (1.0017–1.0097) | 1.007 | 0.001* | (1.0029–1.0105) |
| | Ice cream (frozen desserts) | 1.011 | 0.015 | (1.0022–1.0203) | 1.010 | 0.069 | (0.9993–1.0201) | 1.011 | 0.016* | (1.0021–1.0201) | 1.005 | 0.070 | (0.9992–1.0199) |

[a]Weighted using sample weight [wtfa_sa]. Logistic regression with IBD as outcome; Data source: Sample Adult Cancer file from 2015 NHIS Data release source (https://www.cdc.gov/nchs/nhis/nhis_2015_data_release.htm)

[b]Additional details in survey questions can be found in NHIS 2015 Data release website: ftp://ftp.cdc.gov/pub/Health_Statistics/NCHS/Dataset_Documentation/NHIS/2015/cancerxx_layout.pdf

[c]Food groups are based on the relationship previously established according the dietary guidelines. Details can be found on https://epi.grants.cancer.gov/nhanes/dietscreen/relationship.html.

[d]Vegetables other than lettuce salads, potatoes, cooked beans in which participant already answered to in previous questions.

[e]Food items in this group exclude artificially sweetened or sugar-free kinds

[f]Each food item adjusted for demographic factors: Age, race, poverty status, sex, ethnicity, region

[g]Each food item adjusted for lifestyle factors: Smoking, alcohol user status, alcohol consumption rate, BMI

[h]Odds of having IBD with every unit increase in consumption of respective food item. Shown in 3 decimal digits to show its place within 95% CI.

*Statistically significant; Below the significance level of 0.05

**Comparison of average monthly consumption rate and prevalence of IBD.** Next, we characterized the monthly average (median, 50th percentile) consumption for each food items. Average monthly consumption rate has been determined for the estimated US adult population (**S7 Table**). Using the average value identified in the weighted analysis as a cut off value, we further stratified IBD and non-IBD population into two sub groups (**Table 3**): 1) Monthly consumption greater than the average (> median); 2) Monthly consumption less than or equal to average (≤ median). In an assessment of food intake in the past 30 days, greater estimated number of people with IBD consumed 100% pure fruit juice below the average compared to non-IBD population (55.17% vs. 47.75%, respectively, **Table 3**). Furthermore, more people with IBD consumed cheese (53.05% vs 43.88%) and cookies (53.25% vs 46.46%) at a greater than average rate than the general non-IBD population (**Table 3**). Comparison of the point prevalence of IBD in each consumption rate strata (**S8 Table**) has found the significance to parallel the findings just mentioned. In brief, prevalence of IBD was significantly higher in group drinking 100% pure fruit juice below the average rate (Point prevalence = 1.4892, 95% CI [1.487–1.491]), but lower in group that consumed cheese (Point prevalence = 1.0833, 95% CI [1.082–1.085]) and cookies (Point prevalence = 1.4892, 95% CI [1.487–1.491]) less often.

To determine the odds of having been told of IBD diagnosis in population whose consumption is less than or equal to the average rate compared to those who consumed beyond the average monthly intake, we applied logistic regression. After adjusting for covariates in a single model, we found people eating cheese (OR = 0.629, 95% CI [0.491–0.807], p-value<0.0001) or sport and energy drinks (OR = 0.665, 95% CI [0.493–0.896], p-value = 0.008) at or below the average to be less likely to have been told that they have IBD (**S9 Table**).

**Table 3. Comparison in proportions of estimated subpopulation (w/ or w/out IBD) with different average (Median) consumption frequency, NHIS 2015[a,b].**

| Food groups[c] | Food items | Monthly median[b] | IBD = Yes > Median N | IBD = Yes > Median Percent | IBD = Yes < = Median N | IBD = Yes < = Median Percent | IBD = No > Median N | IBD = No > Median Percent | IBD = No < = Median N | IBD = No < = Median Percent | Adjusted Wald Test[g] F-test | Adjusted Wald Test[g] P-value |
|---|---|---|---|---|---|---|---|---|---|---|---|---|
| | | Weighted | Weighted, Unadjusted | | | | Weighted, Unadjusted | | | | | |
| Whole wheat grains | Popcorn | 1.00 | 955,904 | 32.60 | 1,976,132 | 67.40 | 76,826,625 | 34.37 | 146,696,044 | 65.63 | 0.28 | 0.5987 |
| | Cereal (hot or cold)[h] | 5.00 | 1,392,172 | 47.70 | 1,526,191 | 52.30 | 106,469,140 | 47.35 | 118,408,725 | 52.65 | 0.01 | 0.9175 |
| | Brown rice[f] | 1.00 | 1,151,428 | 39.27 | 1,780,608 | 60.73 | 95,612,663 | 42.70 | 128,317,454 | 57.30 | 1.25 | 0.2643 |
| | Whole grain bread | 8.67 | 1,571,509 | 53.60 | 1,360,527 | 46.40 | 109,705,733 | 49.08 | 113,821,358 | 50.92 | 2.12 | 0.1465 |
| Fruits and vegetables | Fries | 4.33 | 1,583,515 | 54.26 | 1,334,654 | 45.74 | 116,264,228 | 51.85 | 107,984,244 | 48.15 | 0.55 | 0.4600 |
| | Salad (green leafy, lettuce) | 13.00 | 1,107,710 | 37.78 | 1,824,326 | 62.22 | 88,256,583 | 39.32 | 136,190,405 | 60.68 | 0.22 | 0.6375 |
| | Fruit juices (100% pure fruit juice) | 4.33 | 1,315,828 | 44.83 | 1,619,624 | 55.17 | 117,252,424 | 52.25 | 107,142,170 | 47.75 | 4.78 | 0.0295* |
| | Vegetables[d] | 21.67 | 1,441,793 | 49.17 | 1,490,243 | 50.83 | 110,307,851 | 49.28 | 113,536,998 | 50.72 | 0.00 | 0.9761 |
| | Potato (non-fried) | 4.33 | 1,796,601 | 61.27 | 1,135,435 | 38.73 | 133,460,620 | 59.59 | 90,497,203 | 40.41 | 0.26 | 0.6104 |
| | Fruits (fresh, frozen, canned) | 21.67 | 1,308,944 | 44.70 | 1,619,409 | 55.30 | 103,834,256 | 46.26 | 120,604,824 | 53.74 | 0.22 | 0.6376 |
| | Pizza (frozen, fast food, homemade)[h] | 2.00 | 1,148,250 | 39.16 | 1,783,786 | 60.84 | 96,845,835 | 43.22 | 127,230,576 | 56.78 | 1.58 | 0.2099 |
| | Tomato sauce | 3.00 | 1,408,009 | 48.08 | 1,520,344 | 51.92 | 104,850,255 | 46.90 | 118,715,851 | 53.10 | 0.15 | 0.6996 |
| | Salsa (made with tomatoes) | 2.00 | 1,332,345 | 45.44 | 1,599,691 | 54.56 | 99,529,375 | 44.47 | 124,268,240 | 55.53 | 0.09 | 0.7640 |
| | Beans | 4.00 | 1,390,076 | 47.41 | 1,541,960 | 52.59 | 111,421,107 | 49.78 | 112,414,114 | 50.22 | 0.54 | 0.4642 |
| Dairy | Milk (cow milk, any type) | 13.00 | 1,167,197 | 39.87 | 1,760,330 | 60.13 | 99,933,350 | 44.48 | 124,749,996 | 55.52 | 2.11 | 0.1472 |
| | Cheese (excludes cheese on pizza) | 13.00 | 1,553,612 | 53.05 | 1,374,741 | 46.95 | 98,156,072 | 43.88 | 125,524,828 | 56.12 | 8.78 | 0.0033* |
| | Pizza (frozen, fast food, homemade)[h] | 2.00 | 1,148,250 | 39.16 | 1,783,786 | 60.84 | 96,845,835 | 43.22 | 127,230,576 | 56.78 | 1.58 | 0.2099 |
| | Ice cream (frozen desserts)[h] | 2.00 | 1,467,104 | 50.04 | 1,464,932 | 49.96 | 100,845,000 | 45.13 | 122,631,600 | 54.87 | 2.28 | 0.1318 |
| Meat | Processed meat | 4.33 | 1,582,027 | 54.02 | 1,346,326 | 45.98 | 118,598,914 | 53.02 | 105,091,888 | 46.98 | 0.10 | 0.7496 |
| | Red meat | 8.67 | 1,315,018 | 44.91 | 1,613,335 | 55.09 | 103,391,312 | 46.22 | 120,295,901 | 53.78 | 0.18 | 0.6750 |
| Sweetened food/ drinks[e] | Cereal (hot or cold)[h] | 5.00 | 1,392,172 | 47.70 | 1,526,191 | 52.30 | 106,469,140 | 47.35 | 118,408,725 | 52.65 | 0.01 | 0.9175 |
| | Cookies (i.e. cake, pies, brownies) | 3.00 | 1,561,285 | 53.25 | 1,370,751 | 46.75 | 103,811,055 | 46.46 | 119,620,113 | 53.54 | 4.18 | 0.0419* |
| | Donut (i.e. Danish, pastries, muffins) | 1.00 | 1,206,191 | 41.14 | 1,725,845 | 58.86 | 95,378,659 | 42.66 | 128,188,042 | 57.34 | 0.24 | 0.6213 |
| | Candy (i.e. chocolates) | 4.33 | 1,729,593 | 58.99 | 1,202,443 | 41.01 | 125,187,020 | 56.02 | 98,275,868 | 43.98 | 0.86 | 0.3540 |
| | Sports and energy drinks[f] | 0.00 | 865,600 | 29.52 | 2,066,436 | 70.48 | 63,591,743 | 28.30 | 161,101,585 | 71.70 | 0.20 | 0.6544 |
| | Coffee or tea (sugar or honey added) | 4.33 | 1,399,154 | 47.72 | 1,532,882 | 52.28 | 112,756,104 | 50.21 | 111,829,983 | 49.79 | 0.62 | 0.4308 |
| | Fruit drinks (sweetened with sugar)[f] | 0.00 | 860,899 | 29.36 | 2,071,137 | 70.64 | 62,814,773 | 27.98 | 161,702,108 | 72.02 | 0.28 | 0.5956 |
| | Regular soda or pop | 2.00 | 1,325,501 | 45.10 | 1,613,634 | 54.90 | 102,345,627 | 45.54 | 122,382,694 | 54.46 | 0.02 | 0.8899 |

(*Continued*)

**Table 3.** (Continued)

| Food groups[c] | Food items | Weighted Monthly median[b] | Weighted, Unadjusted IBD = Yes > Median N | Percent | < = Median N | Percent | Weighted, Unadjusted IBD = No > Median N | Percent | < = Median N | Percent | Adjusted Wald Test[g] F-test | P-value |
|---|---|---|---|---|---|---|---|---|---|---|---|---|
| | Ice cream (frozen desserts)[h] | 2.00 | 1,467,104 | 50.04 | 1,464,932 | 49.96 | 100,845,000 | 45.13 | 122,631,600 | 54.87 | 2.28 | 0.1318 |

[a]Weighted using sample weight [wtfa_sa]; Data source: Sample Adult Cancer file from 2015 NHIS Data release source (https://www.cdc.gov/nchs/nhis/nhis_2015_data_release.htm)

[b]Unit: Times per month; Additional details in survey questions can be found in NHIS 2015 Data release website: ftp://ftp.cdc.gov/pub/Health_Statistics/NCHS/Dataset_Documentation/NHIS/2015/cancerxx_layout.pdf

[c]Food groups are based on the relationship previously established according the dietary guidelines. Details can be found on https://epi.grants.cancer.gov/nhanes/dietscreen/relationship.html.

[d]Vegetables other than lettuce salads, potatoes, cooked beans in which participant already answered to in previous questions.

[e]Food items in this group excludes artificially sweetened or sugar-free kinds

[f]The median for these diet items are 0, or none. Equivalent to having never consumed in past month.

[g]Test of proportion compares the proportion of IBD population eating > Median (or ≤Median) to the non-IBD population eating > Median (or ≤Median)

[h]Food items appear in more than one food groups: Pizza, Ice cream, Cereal

*Statistically different; Below the significance level of 0.05

**Association of monthly food intake and IBD prevalence: Those eating more than average.** To identify the association of different average monthly food consumption and the odds of having been told of IBD diagnosis, we modeled IBD with overall consumption frequency in logistic regression for the subpopulation eating 1) greater than the average (> median) and 2) at or below the average (≤ median) monthly intake. In an unadjusted model (**S10 Table of S10.1 Table**), in those who are already consuming greater than the average monthly rate, increasing the vegetable intake was associated with slightly smaller likelihood of ever been told one has IBD (OR = 0.98, 95% CI [0.9621–0.9948], p-value = 0.010, **S10 Table of S10.1 Table**). Increasing the intake of regular soda in those who already drink more than the average rate was associated with higher likelihood of having been told of IBD diagnosis (OR = 1.01, 95% CI [1.0038–1.0111], p-value< 0.001, **S10 Table of S10.1 Table**). The effect of eating more vegetable and regular soda more frequently in this population remained unaltered even after adjusting for demography and lifestyle (**Table 4**). Interestingly, increasing the intake of fries in those who are already eating more than the average rate had no significant change in the IBD odds (p-value = 0.380, **S10 Table of S10.1 Table**). In addition, further increasing the intake of non-fried potatoes, ice cream, or coffee or tea in this population was associated with statistically significant, but slightly higher likelihood of having been told of IBD diagnosis (**S10 Table of S10.1 Table**). Adjusting for demography or demography and lifestyle both negated this effect of non-fried potatoes, ice cream, and coffee or tea on the likelihood (**S10 Table of S10.2 Table**).

**Association of monthly food intake and IBD prevalence: Those eating at or below the average.** In an unadjusted model, in population who reported eating at or below the average monthly rate of popcorn or whole grain bread, an increase in the intake of those foods was associated with lower likelihood of having been told of IBD diagnosis (OR = 0.62, 95% CI [0.4290–0.9087], p-value = 0.014) and OR = 0.91 (95% CI [0.8606–0.9679], p-value = 0.002), respectively (**S10 Table of S10.1 Table**). However, increasing the intake of fries resulted in

**Table 4. Association (OR[h]) of IBD and increasing consumption frequency in US subpopulation of >median or ≤median eating pattern (Adjusted for demography and lifestyle), NHIS 2015[a,b].**

| Food groups[c] | Food items | Weighted, Adjusted for Demography and Lifestyle | | | Weighted, Adjusted for Demography and Lifestyle | | |
|---|---|---|---|---|---|---|---|
| | | Consumption Rate > Median | | | Consumption Rate ≤ Median | | |
| | | OR | p-value | 95% CI | OR | p-value | 95% CI |
| Whole wheat grains | Popcorn | 0.97 | 0.135 | (0.9404–1.0084) | 0.68 | 0.041* | (0.4642–0.9837) |
| | Cereal (hot or cold)[i] | 1.00 | 0.660 | (0.9942–1.0092) | 1.08 | 0.122 | (0.9799–1.1862) |
| | Brown rice | 1.01 | 0.493 | (0.9899–1.0213) | 1.01 | 0.972 | (0.5499–1.8581) |
| | Whole grain bread | 0.98 | 0.135 | (0.9656–1.0048) | 0.92 | 0.005* | (0.8605–0.9738) |
| Fruits and vegetables | Fries | 1.01 | 0.610 | (0.9972–1.0167) | 1.23 | 0.001* | (1.0897–1.3941) |
| | Salad (green leafy, lettuce) | 1.00 | 0.821 | (0.9895–1.0133) | 0.97 | 0.085 | (0.9349–1.0043) |
| | Fruit juices (100% pure fruit juice) | 1.00 | 0.721 | (0.9943–1.0084) | 1.11 | 0.228 | (0.9352–1.3231) |
| | Vegetables[d] | 0.98 | 0.012* | (0.9628–0.9952) | 1.01 | 0.704 | (0.9759–1.0367) |
| | Potato (non-fried) | 1.01 | 0.164 | (0.9968–1.0192) | 1.07 | 0.410 | (0.9051–1.2756) |
| | Pizza (frozen, fast food, homemade)[i] | 0.99 | 0.767 | (0.9437–1.0437) | 1.13 | 0.314 | (0.8923–1.4240) |
| | Fruits (fresh, frozen, canned) | 1.00 | 0.555 | (0.9898–1.0055) | 1.01 | 0.580 | (0.9817–1.0335) |
| | Tomato sauce | 1.01 | 0.437 | (0.9884–1.0274) | 0.98 | 0.863 | (0.8021–1.2030) |
| | Salsa (made with tomatoes) | 0.99 | 0.312 | (0.9649–1.0115) | 0.95 | 0.672 | (0.7380–1.2165) |
| | Beans | 1.00 | 0.882 | (0.9903–1.0114) | 1.04 | 0.637 | (0.8899–1.2098) |
| Dairy | Milk (cow milk, any type) | 1.00 | 0.871 | (0.9890–1.0094) | 1.00 | 0.944 | (0.9574–1.0414) |
| | Cheese (excludes cheese on pizza) | 1.00 | 0.628 | (0.9916–1.0051) | 1.02 | 0.413 | (0.9766–1.0593) |
| | Pizza (frozen, fast food, homemade)[i] | 0.99 | 0.767 | (0.9437–1.0437) | 1.13 | 0.314 | (0.8923–1.4240) |
| | Ice cream (frozen desserts)[i] | 1.01 | 0.215 | (0.9958–1.0189) | 0.90 | 0.403 | (0.6929–1.1595) |
| Meat | Processed meat | 1.00 | 0.869 | (0.9862–1.0118) | 1.14 | 0.146 | (0.9563–1.3484) |
| | Red meat | 1.00 | 0.446 | (0.9824–1.0079) | 1.04 | 0.198 | (0.9815–1.0940) |
| Sweetened food/drinks[e] | Cereal (hot or cold)[i] | 1.00 | 0.660 | (0.9942–1.0092) | 1.08 | 0.122 | (0.9799–1.1862) |
| | Cookies (i.e. cake, pies, brownies) | 1.00 | 0.782 | (0.9886–1.0154) | 0.99 | 0.945 | (0.8220–1.2006) |
| | Donut (i.e. Danish, pastries, muffins) | 1.01 | 0.059 | (0.9996–1.0226) | 1.43 | 0.081 | (0.9562–2.1354) |
| | Coffee or tea (sugar or honey added) | 1.00 | 0.187 | (0.9986–1.0070) | 1.12 | 0.319 | (0.8952–1.4034) |
| | Fruit drinks (sweetened with sugar) | 0.99 | 0.468 | (0.9804–1.0092) | n.a | n.a | n.a |
| | Candy (i.e. chocolates) | 1.00 | 0.244 | (0.9969–1.0122) | 1.02 | 0.810 | (0.8564–1.2192) |
| | Sports and energy drinks | 0.99 | 0.375 | (0.9768–1.0089) | n.a | n.a | n.a |
| | Regular soda or pop | 1.01 | <0.001* | (1.0035–1.0110) | 1.12 | 0.455 | (0.8341–1.4976) |
| | Ice cream (frozen desserts)[i] | 1.01 | 0.215 | (0.9958–1.0189) | 0.90 | 0.403 | (0.6929–1.1595) |

[a]Weighted using sample weight [wtfa_sa]. Logistic regression with IBD as outcome; Data source: Sample Adult Cancer file from 2015 NHIS Data release source (https://www.cdc.gov/nchs/nhis/nhis_2015_data_release.htm)

[b]Additional details in survey questions can be found in NHIS 2015 Data release website: ftp://ftp.cdc.gov/pub/Health_Statistics/NCHS/Dataset_Documentation/NHIS/2015/cancerxx_layout.pdf

[c]Food groups are based on the relationship previously established according the dietary guidelines. Details can be found on https://epi.grants.cancer.gov/nhanes/dietscreen/relationship.tml.

[d]Vegetables other than lettuce salads, potatoes, cooked beans in which participant already answered to in previous questions.

[e]Food items in this group excludes artificially sweetened or sugar-free kinds

[f]Each food item adjusted for demographic factors: Age, race, poverty status, sex, ethnicity, region

[g]Each food item adjusted for lifestyle factors: Smoking, alcohol user status, alcohol consumption rate, BMI

[h]Odds of having IBD with every unit increase in consumption of respective food item in the subgroup consuming either > Median or ≤Median

[i]Food items appear in more than one food groups: Pizza, Ice cream, Cereal

n.a: The median for these diet items are 0, or none. Equivalent to having never consumed in past month.

*Statistically significant; Below the significance level of 0.05

higher likelihood of having been told of IBD diagnosis by 19% (95% CI [1.0511–1.3433], p-value = 0.006; **S10 Table** of S10.1 Table). Such effects remained significant even after adjusting for both demography and lifestyle variables (**Table 4**).

## Discussion

We evaluated current US adults affected by IBD and their estimated intake of food items using the nationally representative datasets from NHIS 2015. Our analysis sustains the trend of IBD prevalence in men and women as previously reported [1,41,42]. It is also important to note the finding of overrepresentation of White in disease prevalence to reflect potential disease susceptibility or health disparity due to racial and genetic differences. However, it is also likely for the information to reflect underestimation or under-representation of disease diagnosis in racial minorities attributable to differences in health equity [43,44].

In our assessment of behaviors associated with health risks, compared to the non-IBD population, we found people who have ever smoked to be more prevalent in IBD population. While a greater proportion of IBD population was identified as current drinkers than former drinkers, this percentage was significantly less compared to the population without IBD. Instead, we found that people with IBD were more likely to have been a former drinker (25.68%) than the non-IBD general population (14.06%). Moreover, population with IBD were also more likely to drink infrequently than the rest of the non-IBD population. While IBD was least prevalent in underweight (BMI < 18.5) population, differential distribution of all levels of BMI in population with IBD was unremarkable from the distribution found in the general population without IBD. Diagnosis of chronic disease often induces changes in health-related behaviors such as smoking, drinking, substance use, physical activity, and diet [45–47]. Cross-sectional and longitudinal studies of US men and women diagnosed with chronic illnesses report reduction in smoking [45,46] and drinking [45,46,48] following the diagnosis. Health behavioral change over time was also greater in cohort with disease diagnosis than the healthy controls [45].

We found several food items more likely to be consumed by people with IBD than people without IBD (**Fig 4**). Of 26 food items assessed, greater proportion of the estimated IBD population to ate fries in the past 30 days than the non-IBD population, but consumed other foods similarly in frequency. Popcorn consumption was also less prevalent in the IBD population, although this finding in the weighted but unadjusted analysis was only marginal. We also assessed the odds of ever being told that a person has IBD with respect to whether he/she has eaten certain food items at least once in the past 30 days. An association was found for fries or sports and energy drinks and higher likelihood of having been told of IBD diagnosis. While the association found for fries was not sensitive to demography and/or lifestyle adjustment, association found for sport and energy drink was affected by the adjustment. This sensitivity can be explained by reportedly a strong influence of age, gender, and poverty level have on the intake of sports and energy drinks [49]. We also saw in the multivariate regression analysis, a slight increase in the likelihood of having been told of IBD diagnosis when increasing the intake of fries, cheese, or regular soda. Compared to the general non-IBD population, people with IBD were also more likely to consume cheese and cookies in higher frequency per month.

Our current study also found the likelihood of having been told one has IBD to be smaller for those who have reported eating milk or popcorn in the past month. This observation was also seen in those who drank greater than the average monthly frequency of 100% pure fruit juice. In addition, consuming cheese less than or equal to monthly average frequency or not having consumed sport and energy drink during the past month was associated with smaller

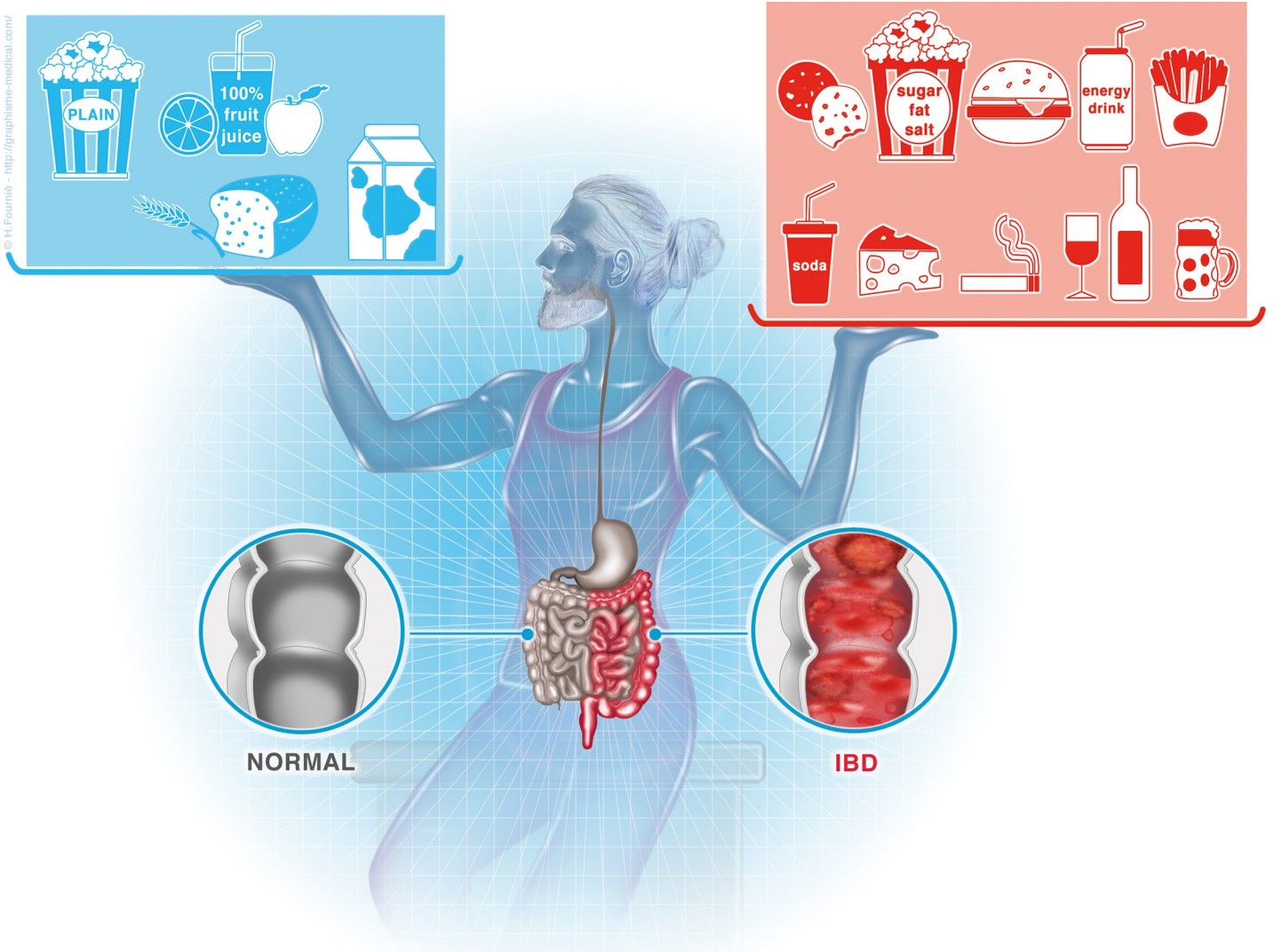

**Fig 4. Simple illustration of food items found to be associated with IBD prevalence in estimated US adult population from the assessment of NHIS 2015.** This figure does not imply or establish causal relationship of food items assessed with IBD. The picture only depicts general association found in this current study.

odds of having been told of IBD diagnosis compared to those whose intake was greater, regardless of demography and lifestyle background.

In population eating food item at a frequency at or below the average, we wanted to see if changing their current eating frequency pattern by increasing the intake would affect their likelihood of having IBD. This analysis was done by regression modeling. After accounting for covariates, we found greater odds of having been told of IBD diagnosis with an increase in the intake of fries in those who eat less frequently than the monthly average. However, this observation was not seen in the population whose intake was already above average. Also, increasing the intake of popcorn was associated with lower odds of having been told of IBD diagnosis in a population whose consumption was already less, but the association was absent in population whose intake was greater than the average monthly rate. While increasing the intake of regular soda among those who already drink more than the monthly average frequency was positively associated with higher likelihood of having been told of IBD diagnosis, this association was not

seen among people who originally reported eating at or below the average. This selective effect was also seen with the vegetable consumption, where increasing the intake of vegetable among those who already consume more than the monthly average was associated with lower odds of having been told of IBD diagnosis. This association was not present in people whose consumption was reportedly below the monthly average.

NHIS 2015 has included Dietary Screener Questionnaire (DSQ) developed by the Risk Factor Assessment Branch of National Institutes of Health, National Cancer Institute, Division of Cancer Control & Population Sciences [50]. Food items were included in the survey as a part of diet and nutrition assessment of CCM. In the present analysis of a national survey of over 33,600 sample US adults, the lack of finding the breadth of differential consumption pattern in our estimated IBD and non-IBD population was surprising. However, we did see some similarity and differences with other literature in the assessment of dietary intake among cohorts with IBD. Our analysis has shown a marginal decrease in the popcorn intake in IBD population compared to the non-IBD population. Intake of dietary fiber has shown to benefit the gastrointestinal symptom management of clinical IBD and improvement in immune response and intestinal lesions in experimental colitis animal model [51]. Likewise, popcorn is a part of whole grain food group, a source of dietary fiber [52,53]. However, several studies suggest high avoidance of popcorn in people with IBD due to its adverse effect on gastrointestinal flares [28,54], in which it was also noted that the people with IBD would normally eat popcorn, just not during the flare-ups [28].

Our analysis has found a positive association between having consumed fries and odds of having been told one has/had IBD based on the food assessment in our study population with and without IBD. Fries consumption was also more prevalent in IBD population compared to non-IBD population but saw no difference in the monthly intake frequency between the two populations. In contrast, Vagianos et al. found intake of fried potatoes to be significantly less in the IBD cohort than matched controls [28] and fried foods in general were perceived by IBD patients to worsen their symptoms [54]. The health risk associated with fries or fried food consumption has been reported by many studies [55–59]. A recent publication has associated high intake of fried potatoes, but not unfried potatoes, with an increase in 8-year mortality rate [55]. However, the study was done in patients with high risk for osteoarthritis who may have underlying medical conditions [55]. Potato is a vegetable that can be a nutritious part of a balanced diet and is abundant in micronutrients such as potassium, vitamin C, vitamin B6, folate, phosphorous, calcium, magnesium, niacin, and others that can assist in meeting daily recommendation of nutrients [60,61]. Studies suggest the food processing such as frying in oil, but not necessarily the food itself, to offset the impact of food on health [56,62–64] and the nutrient quality [62,65–67].

Odds of having been told one has/had IBD was also influenced by different frequency intakes of cheese, regular soda, sports and energy drinks, milk, and 100% pure fruit juice in our analysis. Cheese consumption was more prevalent in IBD population and we found consumption to also increase the likelihood of having been told of IBD diagnosis. While several studies have noted milk and milk products to exacerbate gastrointestinal symptoms and thus avoided by IBD patients [28,54], a systemic review on the effect of milk/dairy product on chronic inflammatory disorders has found the consumption to be beneficial [68] with the effect of cheese being inconsistent and yogurt alleviating the symptoms [26–28,54]. There is limited evidence on the impact of consumption of sport and energy drinks, carbonated beverages such as soda and pop, and 100% fruit juice on IBD risk or symptom management. We found an intake of regular soda and sports and energy drinks to be very prevalent in IBD population in our analysis, which is consistent with other findings [26–28,48] but in disagreement with the findings of Cohen et al [54] where the comparison to non-IBD population was not

made. Nonetheless, consumption of high-sugar, as found in soda and regular sport and energy drinks, was found to be positively associated with increased odds of CD and UC development [69]. High intake of total sugar or in the form of sucrose [30] or monosaccharide [70] was also significantly associated with increased odds of developing IBD [70], but more profoundly in UC than CD [30].

Consideration of diet as one of the etiological factors in IBD has been secondary to a more established link between the disease and adverse immune response [8–11]. Nonetheless, the understanding of the critical influence the environmental factors such as diet, lifestyle, and social factors have on IBD pathology has grown tremendously [8,13,71]. Now, there is increasing evidence that suggests a critical interaction between diet and microbiota as another etiological factor in IBD development. While the elaboration of the role of gut microbiota in IBD is outside the scope of this paper, it is worth mentioning a potential influence diet may have on maintaining the stable expression of human gut microbiota critical for general health and nutrient metabolism [72,73]. Recent studies and reviews have begun to suggest a potential contribution of diet to the microbiota density variation in the human gut and obesity-related health outcomes [74–78]. A review by Viennois et al. suggests that there may be a potential gain in adjusting the diet to accommodate the variability of microbiota composition that exists within population and the composition unique to certain inflammatory diseases, including IBD [78]. Different metabolic by-products made available by diverse dietary patterns and foods introduced to the host is thought to be critical in the maintenance of the homeostatic microbiota composition throughout gastrointestinal tract [74]. As the pathogenesis of IBD is associated with persistent inflammation present in the digestive tract, it is then conceivable to speculate the role of dysregulation of gut microbiota in IBD development. Indeed, numerous literature reviews emphasize the importance of considering diet as one of the environmental factors driving the microbiome environment, or "dysbiosis" that increases the risk for gut inflammatory response [27,79–81], including "westernized" diets that are high in protein, fat, sugar, salt, alcohol, but low in fruit and vegetable consumption. Animal studies have demonstrated the consumption of emulsifiers that are widely present in certain food items to elicit low-grade inflammation due to the thinning of the intestine's primary barrier (mucosal layer) and the displacement of gut microbiota [77,82] and to promote the development of colitis and colitis-associated cancer [82,83]. Research linking diet, microbiota, and development of IBD need much elucidation. Nevertheless, based on the role of diet in inflammatory diseases and evidence of its role in other chronic ailments, significant disruption to healthy gut microbiota via diet modification and eating habits is a very plausible pathway toward developing IBD mediated by abnormal gastrointestinal activities, including nutrient deficiency, metabolism, and inflammation.

## Limitations

From our analysis, certain health behavioral traits and food intake associated with IBD population can be inferred. The strength of our current study is the utilization of a national health survey in which the estimations representative of US adult population can be made. To our knowledge, current analysis is the first to assess food intake pattern in nationally representative IBD population in the US. However, there are significant limitations to this study. Thus, making inferences and drawing conclusions made from the results must be taken cautiously. NHIS is a cross-sectional survey in which the assessment is performed at a single point in time, making inferences in the causal relationship of food intake and predicting the development of IBD invalid given the lack of longitudinal follow-up or assessment in current study design. Responses to questions in the questionnaire are based on the recall or recollection of

participant's memory about a past event or self, which often can be under- or over-represented [84,85] with temporal dependency [86]. Inclusion of such recalling method is no exception in the current study on the questions regarding the primary outcome of ulcerative colitis/Crohn's disease, as no cross reference was made with an actual medical record, and the primary predictors of food variables. Furthermore, the survey is not designed to ask questions specifically related to IBD conditions; thus, no differentiation between the two forms (UC or CD). The survey does not ask disease-specific questions such as the disease duration or disease activity (remission vs relapse) or whether changes in certain habits such as health-related behaviors and food consumption pattern are due to disease diagnosis. Dietary recommendation and guidelines exist [25,27,87–90], but whether people with IBD follow such dietary plans [91] or whether such changes to diet are monitored by or communicated with health professionals are unclear. As with the change in health-related behaviors in people with chronic diseases [45], modification of diet and nutrition intake to better manage gastrointestinal symptoms is also likely. Indeed, a cross-sectional study suggests a change of diet in the majority of IBD patients is based on their perception of or attitude toward the benefit of the diet in gut symptoms [26]. However, they are also likely to self-direct in their choices of food items than to follow any type of dietary treatments such as low- or high-fiber diet, grain/carbohydrate diet, dairy diet, and low short-chain carbohydrate diet [29]. The complexity in assessing the role of diet in IBD risk or symptom management is also compounded by reports of differential effects of nutrition and diet intervention on the course of disease [87,92–96].

## Conclusion

Our nationally representative assessment of estimated US adults with IBD highlights several demographic and lifestyle factors, and certain food intake and consumption pattern associated with IBD. Current study suggests intake of foods typically perceived as unhealthy as a contributing trait of IBD prevalence in the US, in which the correlation itself is not surprising. It would be important to consider the integration of human biology and continually changing environmental and societal factors for a more comprehensive understanding of IBD risk and pathogenesis. As we saw overall food intake to be similar between the IBD and non-IBD population, the effectiveness of dietary guidelines and its adherence and the limiting factor associated with certain food intake should be evaluated. An ideal study for a better evaluation of the role of diet in IBD would be a longitudinal assessment with a detailed food diary and biomarker measurements before and after the onset of disease symptoms leading up to the IBD diagnosis.

## Supporting information

**S1 Table.**
(DOCX)

**S2 Table. Food items listed in the Diet and Nutrition questionnaire from Cancer Control Supplement, NHIS.**
(DOCX)

**S3 Table. Number of adult survey participants and weighted estimated population consuming listed food items from Cancer Control Supplement, NHIS 2015.**
(DOCX)

**S4 Table. Association (OR) of food consumption and IBD in estimated US population, NHIS 2015.**
(DOCX)

**S5 Table. Association (OR) of food consumption (full model) and IBD in estimated US population, NHIS 2015.**
(DOCX)

**S6 Table. Association (OR) of food consumption frequency (full model) and IBD in estimated US population, NHIS 2015.**
(DOCX)

**S7 Table. Population size and average (median) monthly food intake frequency for US sample adults and estimated adult population, NHIS 2015.**
(DOCX)

**S8 Table. Comparison in point prevalence of colitis in estimated subpopulation with different average (median) monthly food intake rate, NHIS 2015.**
(DOCX)

**S9 Table. Comparison in association (OR) of IBD and different monthly average food intake among estimated US population, NHIS 2015.**
(DOCX)

**S10 Table.**
(DOCX)

## Author Contributions

**Conceptualization:** Moon K. Han, Didier Merlin.

**Data curation:** Moon K. Han.

**Formal analysis:** Moon K. Han.

**Funding acquisition:** Didier Merlin.

**Investigation:** Moon K. Han.

**Methodology:** Moon K. Han, Raeda Anderson.

**Software:** Moon K. Han, Raeda Anderson.

**Supervision:** Emilie Viennois, Didier Merlin.

**Validation:** Raeda Anderson, Emilie Viennois.

**Writing – original draft:** Moon K. Han.

**Writing – review & editing:** Moon K. Han, Emilie Viennois, Didier Merlin.

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
