## [Decision Letter · Decision Letter 0]

4 Mar 2020

PONE-D-19-27229

Examination of food consumption in United States adults and the prevalence of inflammatory bowel disease using National Health Interview Survey 2015

PLOS ONE

Dear Dr. Han,

Thank you for submitting your manuscript to PLOS ONE. After careful consideration, we feel that it has merit but does not fully meet PLOS ONE’s publication criteria as it currently stands. Therefore, we invite you to submit a revised version of the manuscript that addresses the points raised during the review process.

Please address all points raised and specifically comments on self reporting algorithm.

We would appreciate receiving your revised manuscript by Apr 18 2020 11:59PM. To enhance the reproducibility of your results, we recommend that if applicable you deposit your laboratory protocols in protocols.io, where a protocol can be assigned its own identifier (DOI) such that it can be cited independently in the future. For instructions see: http://journals.plos.org/plosone/s/submission-guidelines#loc-laboratory-protocols

We look forward to receiving your revised manuscript.

Kind regards,

Andreas Zirlik, MD

Academic Editor

PLOS ONE

Journal Requirements:

Reviewers' comments:

Reviewer's Responses to Questions

**Comments to the Author**

1. Is the manuscript technically sound, and do the data support the conclusions?

Reviewer #1: Yes

2. Has the statistical analysis been performed appropriately and rigorously? 

Reviewer #1: Yes

3. Have the authors made all data underlying the findings in their manuscript fully available?

Reviewer #1: Yes

4. Is the manuscript presented in an intelligible fashion and written in standard English?

Reviewer #1: Yes

5. Review Comments to the Author

Reviewer #1: The manuscript entitled "Examination of food consumption in United States adults and the prevalence of

inflammatory bowel disease using National Health Interview Survey 2015" by Han et al examines the relationship between food intake and IBD diagnosis using data from the National Health Interview Survey.

Han et al demonstrate that the consumption of certain foods is associated with a history of IBD.

Points:

Major:

1. Since it is known that IBD is a heterogeneous disease and that CD and UC differ in the association with different environmental risk factors (e.g. smoking), could the authors also provide the respective analyses for CD and UC separately? If this is not possible could the authors please specify why this is the case?

2. The authors explain that participants in the survey could account for their food intake per month, per week and per day. It is possible that recollections within smaller time frames are more exact than larger time frames. Could the authors comment on differences between the accounts of participants within different time frames?

3. A large fraction of the discussion section of the manuscript is repetition of the results section. Can the authors please revise the discussion section and shorten the repetitive sections?

Minor:

1. Spelling (large and lower case) is inconsistent within the manuscript (e.g. lines 211,212,242). Can the authors please unify this?

2. English grammar within the results section of the manuscript should be revised.

6. PLOS authors have the option to publish the peer review history of their article (what does this mean?). If published, this will include your full peer review and any attached files.

Reviewer #1: No

---

## [Author Response · Author response to Decision Letter 0]

7 Apr 2020

RE: Manuscript PONE-D-19-27229, Examination of food consumption in United States adults and the prevalence of inflammatory bowel disease using National Health Interview Survey 2015

Dear Dr. Andreas Zirlik, Academic Editor

We would like to thank the editorial board and the reviewers for their thoughtful analysis of our manuscript titled "Examination of food consumption in United States adults and the prevalence of inflammatory bowel disease using National Health Interview Survey 2015". The revised manuscript is enclosed. Our revision centers on the helpful comments of the reviewer. Below, we respond to the reviewers in a point-by-point fashion: 

Reviewer #1: The manuscript entitled "Examination of food consumption in United States adults and the prevalence of inflammatory bowel disease using National Health Interview Survey 2015" by Han et al examines the relationship between food intake and IBD diagnosis using data from the National Health Interview Survey. Han et al demonstrate that the consumption of certain foods is associated with a history of IBD.

Points:

Major:

1. Since it is known that IBD is a heterogeneous disease and that CD and UC differ in the association with different environmental risk factors (e.g. smoking), could the authors also provide the respective analyses for CD and UC separately? If this is not possible could the authors please specify why this is the case?

We thank the reviewer for this insightful question. As the reviewer noted, the two most common forms of IBD are CD and UC. Indeed, the disease characteristics is heterogeneous in many aspects as there are similarities. The question of feasibility of delineating the analysis to CD and UC is very appropriate, given the use of this valuable ascertainment of demographic and health condition information that are population-based and randomly sampled (complex survey design). As highlighted by the reviewer, such delineation would provide differential insight in association between various risk factors and the two forms of IBD, which would be quite valuable. 

In brief, the purpose of NHIS as a part of programs of National Center for Health Statistics, Centers for Disease Control and Prevention, is to gather information on certain health conditions, health-related behaviors, and demographic attributes of non-institutionalized civilians in the United States. Questionnaire on whether the selected participants were ever told by the professionals that they had/have Crohn’s disease or ulcerative colitis were first included in the survey year of 1999 and reintroduced in 2015 and 2016. Unfortunately, the two IBD forms were grouped together (Page 8, lines 175-176, revised version) in one question format and no further questions were asked about the diagnosis of the two separate forms. In addition, neither the questions regarding the symptoms experienced nor the treatments/medicine used by the persons who were told of Crohn’s disease/colitis diagnosis were conducted. As noted in the manuscript (Page 33, lines 554-558, revised version), we have emphasized the weakness of the cross-sectional investigation using NHIS in the limitation section.

2. The authors explain that participants in the survey could account for their food intake per month, per week and per day. It is possible that recollections within smaller time frames are more exact than larger time frames. Could the authors comment on differences between the accounts of participants within different time frames?

We thank the reviewer for this insightful question. Cancer Control Module (CCM) is an added module to the NHIS that was first administered in 1987, but has since evolved as a quinquennial surveillance of cancer prevalence in US adults as a supplement to the NHIS. In the manuscript, we have noted the inclusion (Page 31, lines 507-509, revised version) of Dietary Screener Questionnaire (DSQ) in the CCM, which was developed by the Risk Factor Assessment Branch of National Institutes of Health, National Cancer Institute, Division of Cancer Control & Population Sciences. This very identical DSQ is also used in a more well-known nutrition screening survey, National Health and Nutrition Examination Survey (NHANES 2009-2010, www.epi.grants.cancer.gov/nhis/2015-screener). 

DSQ asks participants “During the past month, how often did you have/eat [food item]? You can tell me per [day week, or month]” (Page 8, line 163). The question is limited to inquire a past 30-day time frame in which the participants would have to recall retrospectively from the day of the interview; this is the only account available. Hence, the results we obtained cannot answer the reviewer’s question of whether we can extrapolate on the differences in the accounts of participants. We clarified this in the revision (Page 7, lines156-157, revised version). 

We thank the reviewer for giving us the opportunity to clarify this point. 

3. A large fraction of the discussion section of the manuscript is repetition of the results section. Can the authors please revise the discussion section and shorten the repetitive sections?

We thank the reviewer for the keen oversight. While the information in the discussion section do seem repetitive due to the coverage of similar risk factors analyzed, we distinguished the content of the Result from the Discussion. In the result section, we emphasize mostly on characterizing the findings of the survey study participants and estimated IBD population. In the discussion section, we compare and contrast the findings we reported for the IBD population to the non-IBD population, thus necessitating the description of the non-IBD population in terms of predictors or risk factors similarly analyzed for the IBD population. Subsequently, we elaborate on the possible reasons for the differences. 

Nevertheless, as suggested by the reviewer, we omitted the redundancy while retaining the information needed for comparing the two different populations. Changes are made throughout the discussion section. 

Minor:

1. Spelling (large and lower case) is inconsistent within the manuscript (e.g. lines 211,212,242). Can the authors please unify this?

We thank the reviewer for this suggestion. The corrections are made throughout the manuscript. 

2. English grammar within the results section of the manuscript should be revised.

We thank the reviewer for this suggestion. We revised the result section with more succinct and correct narratives.

---

## [Decision Letter · Decision Letter 1]

9 Apr 2020

Examination of food consumption in United States adults and the prevalence of inflammatory bowel disease using National Health Interview Survey 2015

PONE-D-19-27229R1

Dear Dr. Han,

We are pleased to inform you that your manuscript has been judged scientifically suitable for publication and will be formally accepted for publication once it complies with all outstanding technical requirements.

With kind regards,

Andreas Zirlik, MD

Academic Editor

PLOS ONE

Additional Editor Comments (optional):

Reviewers' comments:

Reviewer's Responses to Questions

**Comments to the Author**

1. If the authors have adequately addressed your comments raised in a previous round of review and you feel that this manuscript is now acceptable for publication, you may indicate that here to bypass the “Comments to the Author” section, enter your conflict of interest statement in the “Confidential to Editor” section, and submit your "Accept" recommendation.

Reviewer #1: All comments have been addressed

2. Is the manuscript technically sound, and do the data support the conclusions?

Reviewer #1: Yes

3. Has the statistical analysis been performed appropriately and rigorously? 

Reviewer #1: Yes

4. Have the authors made all data underlying the findings in their manuscript fully available?

Reviewer #1: Yes

5. Is the manuscript presented in an intelligible fashion and written in standard English?

Reviewer #1: Yes

6. Review Comments to the Author

Reviewer #1: All comments have been thoughtfully addressed.

I have no further comments and thank the authors for the kind answers.

7. PLOS authors have the option to publish the peer review history of their article (what does this mean?). If published, this will include your full peer review and any attached files.

Reviewer #1: No

---

## [Editor Report · Acceptance letter]

14 Apr 2020

PONE-D-19-27229R1 

Examination of food consumption in United States adults and the prevalence of inflammatory bowel disease using National Health Interview Survey 2015 

Dear Dr. Han:

I am pleased to inform you that your manuscript has been deemed suitable for publication in PLOS ONE. Congratulations! Your manuscript is now with our production department. 

With kind regards,

on behalf of

Univ. Prof. Dr. Andreas Zirlik 

Academic Editor

PLOS ONE